# Comparative Genomics of *Legionella pneumophila* Isolates from the West Bank and Germany Support Molecular Epidemiology of Legionnaires’ Disease

**DOI:** 10.3390/microorganisms11020449

**Published:** 2023-02-10

**Authors:** Ashraf R. Zayed, Dina M. Bitar, Michael Steinert, Christian Lück, Cathrin Spröer, Ingrid Brettar, Manfred G. Höfle, Boyke Bunk

**Affiliations:** 1Department of Biomedical Sciences, Faculty of Medicine and Health Sciences, An-Najah National University, Nablus P.O. Box 7, Palestine; 2Department of Vaccinology and Applied Microbiology, Helmholtz Centre for Infection Research (HZI), Inhoffenstrasse 7, 38124 Braunschweig, Germany; 3Microbiology Research Laboratory, Faculty of Medicine, Al-Quds University, Abu-Dies, East Jerusalem 9993100, Palestine; 4Institut für Mikrobiologie, Technische Universität Braunschweig, 38106 Braunschweig, Germany; 5Medical Faculty Carl Gustav Carus, Institute for Virology, Technische Universität Dresden, 01307 Dresden, Germany; 6Medical Faculty Carl Gustav Carus, Institute for Medical Microbiology and Hygiene, Technische Universität Dresden, 01307 Dresden, Germany; 7Leibniz Institute-German Collections of Microorganisms and Cell Cultures DSMZ GmbH, 38124 Braunschweig, Germany; 8Hildegard-Elisabeth Foundation, 66271 Saarbrücken, Germany

**Keywords:** genomic epidemiology, genotyping, surveillance, molecular ecology

## Abstract

*Legionella pneumophila* is an environmental bacterium and clinical pathogen that causes many life-threating outbreaks of an atypical pneumonia called Legionnaires’ disease (LD). Studies of this pathogen have focused mainly on Europe and the United States. A shortage in *L. pneumophila* data is clearly observed for developing countries. To reduce this knowledge gap, *L. pneumophila* isolates were studied in two widely different geographical areas, i.e., the West Bank and Germany. For this study, we sequenced and compared the whole genome of 38 clinical and environmental isolates of *L. pneumophila* covering different MLVA-8(12) genotypes in the two areas. Sequencing was conducted using the Illumina HiSeq 2500 platform. In addition, two isolates (A194 and H3) were sequenced using a Pacific Biosciences (PacBio) *RSII* platform to generate complete reference genomes from each of the geographical areas. Genome sequences from 55 *L. pneumophila* strains, including 17 reference strains, were aligned with the genome sequence of the closest strain (*L. pneumophila* strain Alcoy). A whole genome phylogeny based on single nucleotide polymorphisms (SNPs) was created using the ParSNP software v 1.0. The reference genomes obtained for isolates A194 and H3 consisted of circular chromosomes of 3,467,904 bp and 3,691,263 bp, respectively. An average of 36,418 SNPs (min. 8569, max. 70,708 SNPs) against our reference strain *L. pneumophila* str. Alcoy, and 2367 core-genes were identified among the fifty-five strains. An analysis of the genomic population structure by SNP comparison divided the fifty-five *L. pneumophila* strains into six branches. Individual isolates in sub-lineages in these branches differed by less than 120 SNPs if they had the same MLVA genotype and were isolated from the same location. A bioinformatics analysis identified the genomic islands (GIs) for horizontal gene transfer and mobile genetic elements, demonstrating that *L. pneumophila* showed high genome plasticity. Four *L. pneumophila* isolates (H3, A29, A129 and L10-091) contained well-defined plasmids. On average, only about half of the plasmid genes could be matched to proteins in databases. In silico phage findings suggested that 43 strains contained at least one phage. However, none of them were found to be complete. BLASTp analysis of proteins from the type IV secretion Dot/Icm system showed those proteins highly conserved, with less than 25% structural differences in the new *L. pneumophila* isolates. Overall, we demonstrated that whole genome sequencing provides a molecular surveillance tool for *L. pneumophila* at the highest conceivable discriminatory level, i.e., two to eight SNPs were observed for isolates from the same location but several years apart.

## 1. Introduction

*Legionella pneumophila* is an environmental bacterium and clinical pathogen causing a severe and often fatal atypical pneumonia called Legionnaires’ disease (LD) [1]. LD is transmitted by inhalation or aspiration of aerosols from contaminated environmental sources. Thus, when a case of LD is diagnosed, other individuals exposed to the same environmental conditions might be at a high risk of becoming infected from the same environmental source. The source of such an outbreak can be determined by molecular characterization of *Legionella* isolates from the patient and from environmental samples [2].

LD is primarily caused by the species *L. pneumophila,* which is widely distributed in natural and man-made freshwater environments. Differentiation of isolates from this species by molecular typing methods is important for identifying the source of outbreak. Many molecular typing methods have been used for epidemiological purposes, including the commonly used method known as Sequence Based Typing (SBT) [3] and Multi-Locus of Variable Number of Tandem Repeats Analysis (MLVA), which has slightly higher discriminatory power [4,5]. However, the discriminatory power of MLVA is still limited by its utility for investigating the intra-clonal level of diversity among outbreaks and often cannot distinguish between outbreak and non-outbreak isolates [6,7]. Nowadays, whole genome sequencing (WGS) has been successfully applied for *L. pneumophila,* providing the highest conceivable discriminatory power to distinguish outbreak from non-outbreak isolates [8,9].

*L. pneumophila* strains Philadelphia, Paris and Lens were the first strains of this species for which the whole genomes had been sequenced completely [10,11]. These *L. pneumophila* strains contain a circular chromosome ranging from 3.3 to 3.5 Mbp, encoding more than 3000 protein coding genes and representing 88% coding density. In fact, the genome of *L. pneumophila* has been shown to be highly dynamic [11] due to Horizontal Gene Transfer (HGT) by mobile elements, plasmids, phages and genomic islands (GIs) [12]. Generally, most WGS studies have focused on outbreak isolates [13,14] if they are not dealing with the genetics and evolution of *L. pneumophila* [15]. Thus, the use of WGS for typing of isolates is generally based on either Single Nucleotide Polymorphism (SNPs) comparison or gene-by-gene comparison [16,17].

Previously, we used MLVA to assess the genotypic characteristics and biogeographical distribution of *L. pneumophila* strains isolated from freshwater systems in hospitals across the West Bank and revealed four clonal complexes, some of which were also prevalent in Germany [5,18,19,20,21]. The aim of our study was to compare representative *L. pneumophila* isolates from two very different geographical areas at the level of their whole genome sequences to reveal their differences across large and small geographical distances, to elucidate their genome dynamics and to understand their genetic make-up for virulence. Therefore, we compared genome sequences from *L. pneumophila* reference strains and our 38 isolates using SNPs comparison. We undertook this analysis based on previously published reference genome sequences of 17 *L. pneumophila* strains (i.e., Lpn-Paris [11], Lpn-Corby [22], Lpn-Philadelphia [10], Lpn-Alcoy 2300/99 [23], Lpn-Lorraine, Lpn-Thunderbay [24], Lpn-D7630, Lpn-D7631, Lpn-D7632, Lpn-Pontiac [25], Lpn-OLDA [25], Lpn-ATCC43290 [26], Lpn-LPE509 [27], Lpn-Lpm7613, Lpn-Lens [11], Lpn-Toronto and Lpn-HL06041035). In addition to whole genome sequencing (WGS) on the Illumina platform, for one environmental isolate from the West Bank (A194) and one environmental isolate from HZI, Germany (H3), we performed complete genome sequencing (CGS) using the Pacific Bioscience (PacBio) platform, leading to completely resolved microbial genomes in term of replicative units—chromosomes and plasmids [28]. Finally, we investigated the genome dynamics of *L. pneumophila* strains by analyzing Genomic islands, mobile elements, and virulence genes of the whole genome sequences.

## 2. Materials and Methods

### 2.1. Selection of L. pneumophila Strains and Reference Genomes

In total, 180 MLVA-8(12) genotyped environmental isolates from the West Bank and 260 clinical and environmental isolates from Germany were used for an initial screening [18]. A subset of 38 isolates was then carefully selected for whole genome sequencing (WGS) based on the following criteria: (i) clonal relatedness based on MLVA-8(12) genotyping using UPGMA cluster analysis; (ii) genotype abundance and (iii) site of isolation. In detail, sixteen environmental isolates from the West Bank were selected. Furthermore, fourteen clinical isolates across Germany and eight environmental isolates from the Helmholtz Center for Infection Research (HZI), Braunschweig, Germany, were sequenced in this study (Appendix A). Additionally, two isolates (A194, H3) were selected for PacBio complete genome sequencing (CGS) for a robust comparison to the short-read-based WGS genomes. Additionally, seventeen *L. pneumophila* genomes representing the broad diversity of the species were retrieved from NCBI GenBank database (https://www.ncbi.nlm.nih.gov/genome/browse#!/prokaryotes/416/, accessed on 30 December 2022) and used as references (Appendix A).

### 2.2. Illumina HiSeq and Pacific Biosciences Genome Sequencing, Assembly and Annotation

Illumina HiSeq 2500 and PacBio *RSII* sequencing technologies were applied in this study. Both forms of sequencing technologies are well established and complementary in terms of read length and error types [29,30]. WGS sequencing was performed on the Illumina HiSeq 2500 platform via the HZI genome analysis unit. In detail, genomic DNA libraries were prepared using the NEBNext Ultra Kit according to the manufacturer’s instructions (NEB, Ipswitch, ML, USA) and sequenced in a 100 bp paired-end run on the Illumina HiSeq 2500 platform. For each genome, approximately 100× coverage was obtained. Processed sequence reads for each isolate were assembled using Velvet version 1.2.10 with a k-mer size of 61. Hereby, sets of contigs were retrieved, but no complete genomes were retrieved.

Complete genome sequencing was performed on the PacBio *RSII* platform at the German Collection for Microorganisms and Cell Cultures (Leibniz Institute DSMZ). Therefore, 5 g weighted cells were grown in YEB medium. Genomic DNA was extracted using Genomic Tip 100/G kit (Qiagen, Hilden, Germany). Subsequently, a SMRTbell^TM^ template library with insert sizes greater than 4 kb was prepared according to the manufacturer’s instructions (Pacific Biosciences, Menlo Park, CA, USA). Processed sequence reads were assembled into complete genomes using HGAP3 (within SMRT Portal 2.3.0) by applying default settings.

Assembled contigs for all sequenced isolates were annotated using the Prokka pipeline v1.11 [31]. Prokka relies on external prediction tools to identify the coordinates of CDS, tRNAs, rRNAs, CRISPRs and other genomic features encoded on contigs and chromosomes. All *L. pneumophila* genomes were annotated using a species-specific database created from the published, manually annotated *L. pneumophila* strain Lpn-Corby [22]. Assembled genome sequence data are available at NCBI GenBank under BioProject accession number PRJNA912116. A detailed list of all GenBank accession numbers is provided in Appendix A. Isolate L10-023 has been reported before at GenBank under BioProject accession number PRJNA278111.

### 2.3. Genome Analyses

For phylogenomic identification and phylogenomic tree construction, GenBank files of 38 *L. pneumophila* clinical and environmental isolates from Germany and the West Bank and 17 reference genomes were submitted to the Type Strain Genome Server (TYGS) (https://tygs.dsmz.de, accessed on 1 November 2022) [32]. The pairwise comparison of the user strains with the *L. pneumophila* Philadelphia1 type strain was performed using GBDP and accurate inter-genomic distances inferred under the “trimming” algorithm and distance formula d4. Digital DNA–DNA hybridization (dDDH) values and confidence intervals were calculated following the recommended settings of GGDC 2.1 [32]. Sub-branching clustering was based on a 93% dDDH threshold, as previously discussed [33]. The tree was rooted at the midpoint analyses [32].

Additionally, the software ParSNP was used for SNP analysis in order to construct a phylogenomic tree for the core genome on the same dataset of the 55 genomes against the closest reference strain Alcoy [33]. ParSNP-generated results were visualized using Gingr [34]. Gingr provides an interactive display of multi-alignment variants and phylogenetic trees estimated from the core genome alignment.

Subsequently, assembled contigs of *L. pneumophila* isolates along with genome sequences of published strains from GenBank database were aligned and visualized using progressive MAUVE algorithm multiple-genome alignment software version 2.4.0 with default parameters. In detail, the fragmented WGS sequences of all genomes have been ordered with the “Move contigs” function within MAUVE to the Alcoy reference. Afterwards, comparative synteny analysis among *L. pneumophila* strains was carried out using MAUVE [35,36].

Genomic island (GI) regions were identified using the web-based tool IslandViewer 4 (http://www.pathogenomics.sfu.ca/islandviewer, accessed on 1 November 2022) [37], which combines the prediction results of three genomic island identification algorithms. Genomic islands present in *L. pneumophila* isolates were predicted by at least one method.

BLASTp was applied to compare the protein sets against the Virulence Factors of Pathogenic Bacteria (VFDB) database [38,39]. Amino acid sequences with a 75% match identity were chosen, and the description of the best hit was assigned as the annotation of the predicted gene compared to the *L. pneumophila* strain Philadelphia as the default bacterium on the webpage.

Clusters of orthologous proteins were generated using ProteinOrtho V5.16b [40]. The core genome was analyzed based on the number of common proteins present in all 55 *L. pneumophila* strains. Paralogous proteins were gathered in the same clusters, and unique genes were calculated for each isolate by applying the “singles” option.

### 2.4. Detection and Identification of Plasmids and Phages

Contigs not belonging to the bacterial chromosome were considered to be plasmids. The FASTA files of these contigs were queried in BLASTn for identification of plasmid-specific genes, such as replication and partitioning genes. Genomes were queried for the presence of phages using PHASTER (PHAge Search Tool Enhanced Release) at (http://phaster.ca, accessed on 1 November 2022 [41]).

## 3. Results

### 3.1. General Features of Genomes from L. pneumophila Isolates

The genomes of 36 *L. pneumophila* genomes were sequenced with approximately 100× coverage using the Illumina HiSeq 2500 platform. A range of 28 to 103 contigs was obtained by Velvet assemblies. Complete genome sequencing of the isolates H3 and A194 yielded two circular chromosomes of 3,587,831 bp and 3,467,904 bp, respectively. Strain H3 additionally comprised a plasmid of 103,432 bp.

Strain H3 is an environmental isolate from the HZI hot water supply. It has been classified as serogroup 6 Chicago, sequence type (ST) 1431 and MLVA genotype Gt14(31). Genomic islands as identified by IslandViewer have an additive total length of 209,867 bp, which represents 5.7% of the genome. A total of 3386 genes, 3336 CDS, 9 rRNA, and 43 tRNA genes were identified. Strain A194 is an environmental isolate from Hebron, West Bank. It has been classified as group 6 Dresden and MLVA genotype Gt40(47). Here, the total size of the genomic islands has been determined to be 156,728bp (4.5%). Its genome comprised 3121 genes: 3071 CDS, 9 rRNA and 43 tRNA genes.

Appendix A summarizes the main features of all *L. pneumophila* isolates and their genomes sequenced by the HiSeq and PacBio platform in comparison to all reference genomes of *L. pneumophila* used in this study.

### 3.2. Phylogenomic and SNP Analysis of L. pneumophila Isolates

Identification of all 55 *L. pneumophila* strains was performed using the TYGS webserver [32]. First, the phylogenomic analysis of our 38 *L. pneumophila* isolates revealed the correct taxonomic affiliation to be *L. pneumophila*. Hereby, dDDH values of 91.3–94.7% were computed against the type strain *L. pneumophila* Philadelphia, fulfilling the criterion for bacterial species identification (Appendix A) [32]. For ParSNP analysis, strain Lpn-Alcoy 2300/99 was shown to be the closest reference strain, with a complete genome available in the GenBank database displaying 71% of the reference shared and 1,930,163 Core-SNPs (Appendix A). Consequently, an SNP-based phylogenomic tree was reconstructed based upon this reference in Figure 1, displaying a more detailed structure than that from TYGS (Appendix A). SNP counts were quantified in Appendix A.

The branches within the tree represent all the major genomic lineages of *L. pneumophila* that were introduced by Gomez-Valero et al. [42], with the exception of branch one, namely Philadelphia included five reference genomes Lpn-LPE509, Lpn-Philadelphila1, Lpn-Thunderbay, Lpn-ATCC 43290 and Lpn-Lpm7613; two clinical isolates from Germany; and two Palestinian strains A156 and A129. Branch two (Lorraine) included only isolate A15 and the reference strain Lpn-Lorraine. The third branch (Lpn-HL060641035) included environmental isolates from Germany (H29, H34 and H35 from HZI as well as outbreak strains from New York City, namely Lpn-D7630, Lpn-D7631 and Lpn-D7632 plus an environmental reference strain from France). Four of our isolates from the West Bank (A139, A5, A131, A29) and one from Germany (H39) clustered in branch four Paris. This branch included the worldwide-distributed sequence type 1 (ST1), which is represented by the reference strains Paris and OLDA, both of which are closely related to our MLVA genotype Gt4(17). A second sub-branch included Gt6(18) isolates A29 and A131, which are found exclusively in Hebron, West Bank, and as an outgroup, the environmental isolate H39 from the HZI, Germany. In branch five (Pontiac), a group of six clinical isolates from Germany clustered together with the reference genome of strain Pontiac.

The highest number of isolates was found in branch six (Alcoy). Two of these isolates (H3 and H23) had been obtained from the same drinking water distribution system (DWSS), but four years apart. Furthermore, cluster six included closely related isolates (all Gt40(47) (L04-545, A193, A194 and A195) from the Warstein outbreak that occurred in Germany during 2013. The third most abundant isolates of MLVA genotype Gt10(93) (A108, A112, A114 and A127) in our collection clustered all together. Lastly, a group of clinical isolates from Germany (L02-465, L06-153, L06-129, L04-041 and L05-341) plus an environmental isolate (A166) from the West Bank clustered together. This highly diverse cluster included two complete reference genomes from LD outbreaks, i.e., strains Alcoy 2300/99 and Corby (Figure 1). Overall, six of the seven phylogenomic branches from the species *L. pneumophila* are covered by isolates from out study [15,43].

### 3.3. Identification of Orthologous Genes in L. pneumophila Isolates

ProteinOrtho analysis identified a core genome of 2367 CDS and an accessory genome of 575 CDS, i.e., the sum of CDS that do not share genes from all our 38 isolates. Hereby, the size of unique accessory genes within each branch varied from 1 (HL06041035) to 177 CDS (Pontiac), as shown in the single petals of the Venn diagram (Figure 2).

Isolates A129 and A156 from the Philadelphia branch have the same MLVA genotype Gt64(74) and branch together in the phylogenomic tree (Figure 1), although A129 contained 90 isolate-specific genes (Appendix A). The majority of those genes are located on a separate contig and contain plasmid-specific genes as a ParB-like partitioning protein, *vir* and *tra* genes and a Toxin/Antitoxin system. Details on isolate-specific genes and their potential functions are provided in Appendix A. For example, the clinical isolate L01-443 from Germany contained two genes encoding metallo-beta lactamase family proteins, which provided antibiotic resistance. Similarly, in the Paris branch, the isolates H39 and A29 possess extrachromosomal genetic elements. In isolate A139, two cell division proteins and a FtsI/penicillin binding protein were detected that confer resistance to antibiotics. In the Pontiac branch, the German clinical isolates L10-091 and L10-023 contained a group of multidrug resistance genes (Appendix A). In the Alcoy branch, isolate A166 contained a set of T4SS genes *icmE/dotG* (Appendix A).

In the Paris branch, a group of 20 genes containing a multidrug resistance ABC transporter ATP-binding protein and protease subunit HflK, which presumably provide antimicrobial resistance, were shared between Gt4(17) and Gt6(18). In the Alcoy branch, the sub-branch of Gt14 shared a group of 35 genes including a multidrug resistance protein MdtK. The sub-branch of the two clinical isolates from Germany that were genotyped as Gt71(135) shared a group of 61 genes, within which were genes encoding heat shock proteins, which might increase the pathogenicity of these isolates.

### 3.4. Comparison of MLVA Genotyping and Whole Genome SNP Analysis

We observed good concordance between the phylogenomic branches and MLVA-genotypes that were determined earlier [18]. Hereby, branch Philadelphia corresponded to MLVA-genotype Gt64(74); branch HL06041035 to Gt22(100) and Gt22(102); branch Paris to Gt4(17) and Gt6(18); branch Pontiac to Gt29(27), Gt69(69), Gt72(66) and Gt75(49); and branch Alcoy to Gt40(47), Gt9(92), Gt10(93), Gt14(40), Gt14(31), Gt71(135), Gt8(132), Gt8(142) and Gt30(137) (Appendix A).

By looking in more detail into the individual branches, it can be seen that Philadelphia contained four isolates, three of which belong to Gt64(74), and the clinical isolate from Germany L09-313 was Gt84(16). The two environmental isolates from the West Bank (A156 and A129) were nearly identical (<4 SNPs), as revealed by whole-genome SNP analysis (Appendix A). However, the clinical isolate L09-313 was highly diverse, comprising 6283 unique SNPs. Isolates of the MLVA genotypes Gt22(100) and Gt22(102) formed a specific sub-lineage but could still be distinguished in most cases. For instance, SNP analysis revealed that some Gt22(100) and Gt22(102) isolates differed by 67 SNPs. On the other hand, in the Paris branch, the genome sequence of isolate A29 differed by only two SNPs from the environmental isolate A131. Interestingly, these isolates were isolated from the same hospital located in Hebron in the Southern West Bank but one and a half years apart (Appendix A). In Paris branch four, the MLVA genotype Gt4(17) formed a specific sub-lineage, but isolates of Gt4(17) from different cities could still be distinguished because they have been isolated in different years. For example, the strains A5 and A139 differed by 119 SNPs. However, the German isolate H39, as it is also Gt4(17), showed a high amount of 11,020 SNPs. Cluster five contained six clinical isolates from Germany. Two of these isolates were Gt75(49), another two were Gt29(27), one was Gt72(66) and the last one was Gt69(69). These cluster five isolates showed a highly different SNP pattern ranging from 823 to 14,949 SNPs among different clinical strains across Germany (Appendix A).

In the Alcoy branch, isolates of Gt40(47) formed MLVA-genotype specific sub-lineages, but closely related outbreak associated isolates from Warstein could be clearly distinguished. For example, SNP analysis revealed that isolates of Gt40(47) that were associated with the Warstein outbreak differed by 76 SNPs. Petzold et al. [44] demonstrated that SNP analysis of targeted genes revealed an average of two SNPs per 100bp. In the isolates from West Bank in hospital H, a higher SNP diversity (2822 SNPs) was identified in comparison to the clinical isolate L04-545. Similarly, isolates of Gt10(93) clustered in one sub-lineage but clearly differed by 541 SNPs. Furthermore, the genome sequence of isolate L06-153, which was Gt71(135), differed by only two SNPs from isolate L06-129 (Figure 1 and Appendix A). Overall, we observed that MLVA genotyping is in good concordance with the SNP genotyping, but it cannot provide as high a resolution on the intra-clonal level as that provided by WGS.

### 3.5. Detailed Analysis of Individual SNPs

An alternative way to evaluate variation among the genomes is to study single nucleotide polymorphism in detail at the level of individual genes. Therefore, we visualized ParSNP results with the software Gingr v 1.0. High-quality SNPs extracted from those positions have SNP changes in targeted genes. As an example of this type of analysis, we used four German isolates of Gt14(30) and Gt14(31), namely H1, H2, H3 and H23, all of which were isolated from the DWSS of the HZI but in different years (2009 and 2013). These isolates differed by just nine SNPs, and most of them were attributed to virulence genes (Appendix A). For example, LpnH3D14_02694 and LpnH3D14_02929 were identified with 96% and 79% similarity by comparison with the Dot/Icm T4SS effector and the phagosome trafficking protein DotA, respectively. These results showed that very closely related *L. pneumophila* strains often harbor alterations in virulence and related genes.

### 3.6. Identification of Mobile Elements in the Alcoy Branch

To investigate structural genomic differences within the Alcoy branch, we used the Illumina Hiseq genome sequences of all twenty-one isolates for comparison with the complete genomes of the reference strains Alcoy 2300/99 and Corby plus the two genome sequences of isolates H3 and A194 that were obtained by the PacBio platform. Multiple whole-genome alignment using Mauve revealed largely syntenic Locally Collinear Blocks (LCBs) (Figure 3A). Notably, clinical and environmental strains from two geographical regions (West Bank and Germany) were included that were highly similar except for two interesting areas. We focused on two different areas that were clearly identified in strains H3, A194 and Lpn-Corby (Figure 3B). Area 1 was an 84 kbp region containing regulatory proteins, antibiotic resistance proteins, genomic islands and three tRNAs (Figure 3B and Appendix A). This area does not exist in the reference genomes of Alcoy 2300/99 and Corby but was clearly present in the isolates H1, H2, H3, H23, A193, A194 and A195. In addition, it was partially present in the isolates L04-041, L05-341, L06-129, L06-153 and A166. The second area with 49 kbp was much smaller and present in the Corby genome as well as in the isolates H1, H2, H3 and H23 (Figure 3). All major gene components of area two are summarized in Appendix A.

### 3.7. Identification and Comparison of Genomic Islands in L. pneumophila Isolates

Looking in more detail into genomic islands (GIs) of branch six isolates revealed high genome plasticity. Three different isolates, A194, H35 and L02-465, gained genes that contribute to adaptation and pathogenicity via heme proteins, cytochrome C proteins, and hemin-binding protein. Isolate A194 gained the 15kb bifunctional hemolysin gene. Clinical isolates L10-091 and L06-129 gained heat shock proteins which aid *L. pneumophila* in improving its virulence by adapting to high body temperatures. In addition, L06-129 gained CRISPR-Cas, which helps the strain to improve its phage defense mechanisms. Isolate A29 contained cold shock protein, which helps it to adapt to cold environments. Strain A112 gained anaerobic sulfatase-maturating enzyme, which helps it to adapt to the environment. The Gt14(30) and Gt14(31) environmental isolates from HZI sub-lineage gained beta-lactamase genes, which render these strains antibiotic resistant (Appendix A).

Overall, it becomes clear that the studied *L. pneumophila* isolates revealed high genome plasticity and adaptation to their environments and pathogenicity by acquisition and loss of genes and genomic islands taken together in the composition of all genomes and up to nine percent of the genome content comprising horizontally transferred genes [15]. A comparison of GIs in *L. pneumophila* branches is provided in (Figure 4, Figure 5 and Appendix A).

### 3.8. L. pneumophila Isolates Encode Proteins from Different Eukaryotic Domains

Genome analysis of all 55 *L. pneumophila* strains revealed the presence of different eukaryotic motifs/domains according to the definition that a eukaryotic domain is one that contains more than 75% of eukaryotic gene sequences (Appendix A). The most abundant eukaryotic domains identified were ankyrin repeats. Hereby, the number of ankyrin repeats varies in the individual branches, e.g., 15–17 within the Paris branch, 12 within Philadelphia and HL060641035 and 11–16 in the Alcoy branch, reflecting a large extension of this branch. Ankyrin motifs are frequently associated with other eukaryotic motifs and, thus, constituted modular proteins associated with eukaryotic F-box and U-box. Notably, in all strains of the Philadelphia branch, three F-boxes and one U-box were detected, whereas in all strains of the Alcoy branch, only one F-box was observed, and no U-box was observed (Appendix A). Clearly, F-box and U-box domains were present in the majority of *L. pneumophila* strains studied, suggesting manipulation of the host ubiquitin-system is a fundamental virulence strategy of *L. pneumophila*.

Protein kinases play an important role in cellular signal transduction. *L. pneumophila* translocate into the host cell kinases that contribute to further signaling modifications. Proper regulation of host-cell signaling by *L. pneumophila* is necessary for its ability to replicate intracellularly while avoiding host defense (Appendix A). Clearly, *L. pneumophila* strains of the Philadelphia and Alcoy branches contained only six to seven kinases, with the exception of the Alcoy sub-branch that comprises the German environmental isolates (H1, 2, 3, 23 and L02-465), which contained eight to ten kinases. Strains of the Paris and Pontiac branches encoded 9 to 13 kinases in addition to strain A15 from the Lorraine branch.

The leucine-rich repeat (LRR) domain is evolutionarily conserved in many proteins associated with innate immunity in eukaryotes. Serving as a first line of defense, the innate immune response is initiated through the sensing of pathogen-associated molecular patterns. Strains from the Philadelphia and Lorraine branch contained five to seven LRR domains. All other *L. pneumophila* branches had only three to five LRR domains (Appendix A).

Genes for cytochrome P450 were observed in all *L. pneumophila* strains, with the exception of the strains from the Philadelphia branch (Appendix A). The SET domain of *L. pneumophila* is a eukaryotic protein motif involved in histone methylation and epigenetic modulation. The SET domain of *L. pneumophila* is involved in the modification of histone H3 in the nucleolus of the host cell, thereby enhancing heterochromatic rDNA transcription. All *L. pneumophila* strains contained one SET and HAD domain (Appendix A). Heat Shock Proteins (HPS) are a family of proteins produced by cells in response to exposure to stressful conditions. They were described in relation to heat shock but are also known to be expressed during other stresses such as cold and UV radiation. Strains from the Paris and Lorraine branch contained only 13 HSP, whereas strains from the Alcoy branch had 14 to 17 HSP (Appendix A).

### 3.9. Identification of Pore-Forming Genes Mediating Cytotoxicity in L. pneumophila Isolates

Central to the pore-forming mediated cytotoxicity of *L. pneumophila* are the *Dot/Icm* loci, which, taken together, directly assemble to a type IV secretion system (T4SS) [45,46]. Furthermore, the toxin RtxA plays an important role in the pore-mediated cytotoxicity [47,48,49]. Although all *L. pneumophila* strains examined until today contain the complete *dot/icm* loci, sequence variations among the *dot/icm* genes among different *L. pneumophila* strains have been reported [45,50]. Eleven *dot/icm* T4SS genes (*icmT* [51,52], *icmS* [53], *icmR* [53], *icmQ* [53], *icmL/dotI* [54], *icmK/dotH* [54], *icmE/dotG* [54], *icmC/dotE* [54], *dotB* [54], *dotA* [54] and *icmW* [53]) and *rtxA* gene [49] are responsible for the pore-forming mediated cytotoxicity of *L. pneumophila*. All of these genes were identified by BLASTp searches against VFDB, based on the inclusion of the *L. pneumophila* strain Philadelphia as the default reference genome. Appendix A and Figure 6 show 75% to 100% identity between the pore-forming mediated cytotoxicity genes. For example, the *icmT* gene shared 82% identity for all strains among twelve VFDB-annotated genes identified from *L. pneumophila* strains. The *icmS* gene was shared by all strains (100%) except for cluster three (99%). For the *icmR* gene, the BLASTp identity varied (from 91 to 97%) for all the strains. The *icmQ* gene had a 100% identity in clusters one, two and six plus a 99% identity for clusters three, four and five. The *icmL/dotI* genes shared 84% identity for all strains on average. The *icmR/dotH*, *icmE/dotG*, *icmC/dotE*, *dotB, dotA* and *icmW* gene shared (79–84%), (87–91%), (99–100%), (99–100%), (78–93%) and (98–100%) identity for all strains, respectively. Moreover, the *rtxA* gene shared (75% to 85%) identity for all *L. pneumophila* strains (Figure 6). Taken together, this detailed analysis suggests that the twelve genes studied out of more than 200 of the *dot/icm* loci of *L. pneumophila* represent a large repertoire of effectors which are necessary for virulence [42,55]. In general, all *L. pneumophila* strains shared the same Dot/Icm T4SS, with less than 25% structural differences at the protein level (Figure 6 and Appendix A).

### 3.10. Identification of Plasmids and Phages in L. pneumophila Isolates

The genome of isolate H3 was completely sequenced using PacBio technology and contained a plasmid, the features of which are summarized in (Appendix A). This plasmid has 111 genes and a GC content of 40%. Most of the genes are accessory genes and commonly found in bacterial plasmids. An important gene on the plasmid was a metallo-beta lactamase family protein (LpnH3D14_03288) because it is involved in the breakdown of antibiotics by antibiotic-resistant bacteria. Additionally, the plasmid carried a cold shock protein (LpnH3D14_03325) which was found to be involved in stress resistance and virulence in bacteria. Moreover, the plasmid contained a group of conjugative elements from the Trb/Tra family (LpnH3D14_03353-71). Furthermore, the isolates A129, A29 and L10-091 carry plasmids consisting of 125, 30 and 37 genes, respectively (Appendix A). In detail, the presumable plasmid of isolate L10-091 carried multi-drug resistance efflux proteins (O6E59_10555, O6E59_10560 and O6E59_10570) which contribute to antibiotic resistance, especially against fluoroquinolones.

All 17 *L. pneumophila* reference strains used in this study and 25 *L. pneumophila* clinical and environmental isolates from Germany and the West Bank contained prophages in their genome. On the other hand, 13 *L. pneumophila* clinical and environmental isolates from Germany and the West Bank did not carry prophages (Appendix A). The length of prophage regions ranged between 5.6 and 70.3 kbp, covering 7 to 29 protein-coding genes. All prophages were designated as incomplete or questionable due to the low completeness scores assigned by PHASTER.

## 4. Discussion

In the present study, we sequenced and compared the genomes of thirty-eight *L. pneumophila* isolates covering different serogroups, MLVA-8(12) genotypes, isolated (clinical and environmental) from the West Bank and Germany. The majority of previous studies have used WGS to study the genomics of *L. pneumophila* and to discriminate outbreak strains of *L. pneumophila* retrospectively or during the outbreak [8,56]. Moran-Gilad el al [56] studied outbreak isolates using core genome multi-locus sequence typing (cgMLST) and observed that cgMLST has a high enough resolution to identify *L. pneumophila* outbreak strains [56]. Another study compared whole-genome SNP analysis with sequence-based typing (SBT) for genotyping *L. pneumophila,* and they found a comparable relationship between genome SNPs and SBT [8]. Qin et al. defined the minimum core genome (MCG) and demonstrated that MCG typing enabled differentiation of *L. pneumophila* strains into groups that have major differences in virulence and phenotypic features [16].

### 4.1. Comparison of Taxonomic Resolution of MLVA Genotyping and SNP Analysis

A threshold value for the number of SNPs necessary to identify outbreak-associated *L. pneumophila* strains still has to be established. For example, more than 200 SNPs have been reported in phylogenetically closely related *L. pneumophila* outbreak strains [13,57]. Mercante et al. [58] showed that up to 20 core SNPs were identified in a comparison of Philadelphia branch *L. pneumophila* isolates. In the present study, up to 120 SNPs were identified in the same MLVA-8(12) genotype from the same sub-lineage in a phylogenomic cluster. In particular, the environmental isolates A156 and A129 which represent Gt64(74) from a biofilm of hospital F in the Bethlehem area were identical for all reference genomes, except that the *L. pneumophila* strain Lens and the *L. pneumophila* strain LPE509 differed by only one and four SNPs, respectively. The environmental isolates H29, H34 and H35 from the HZI differed by less than 70 SNPs. A sub-lineage of cluster six for the clinical isolates L06-153 and L06-129 from Brandenburg, Germany differed by only three SNPs against all reference genomes used in this study. On the other hand, the cluster four sub-lineage of Gt4(17) contained four strains (two reference genomes from the strains Paris and OLDA from France and the U.S.A., respectively, and two environmental isolates A5 and A139 from Jerusalem and Nablus, the West Bank) differed by less than 260 SNPs against all the reference genomes in this study. These results are in concordance with those of Khodr et al. [59]. They sequenced six ST1 genomes (four clinical and environmental isolates from a hospital, and the other two were unrelated) and observed that geographically unrelated isolates differed by more than 1500 SNPs. In comparison, our hospital isolates differed by up to 20 SNPs between two strains. Several previous studies showed that *L. pneumophila* strains with the same genotype (either SBT or MLVA) that are isolated from the same area have less SNP differences than strains isolated from different areas. This demonstrates that the geographical origin is as important as the genotype (Figure 1 and Appendix A).

### 4.2. Analysis of Pore-Forming and Virulence Genes

Morozova et al. [60] showed that the *dot/icm* genes are highly conserved in *L. pneumophila* strains. After whole genome sequencing (WGS) technology became available, Gomez-Valero et al. [42,61] confirmed the findings of the previous study which showed high conservation (98%) among orthologs of the reference strains Lpn-Corby, Lpn-Paris, Lpn-Philadelphia and Lpn-Lens, with few exceptions in the *dotA* gene. The *dotA* gene is an essential gene for virulence activity of *L. pneumophila* strains because it encodes an integral membrane protein with eight domains. This explains why a *dotA* mutant of *L. pneumophila* strain Corby is being used as a negative control for all virulence assays [62]. Costa et al. [63] analyzed 300 *dotA* gene sequences from *L. pneumophila* strains and demonstrated that pathogenic *L. pneumophila* strains belong to a subset of genotypes existing in the environment. Khodr et al. [59] explained the high variation of the *dotA* gene of *L. pneumophila* by indicating that this gene is a target for host speciation and adaptation to different hosts and environments. Dumenil et al. [64] showed that *icmR* is a regulator gene for the *icmQ* gene, which possesses pore-forming activity. In addition, Gomez-Valero et al. [65] demonstrated that *dotB, icmS* and *icmW* are highly conserved genes. These facts are in accordance with our results (Figure 6) which show that *icmR* (91–96%), *icmS* and *dotB* (99–100%) and *icmQ* (98–100%) were highly conserved genes, whereas *dotA* had only a 78% to 93% gene identity for our *L. pneumophila* strains. Overall, the *dot/icm* T4SS is a highly conserved and complex molecular system (Figure 6 and Appendix A).

### 4.3. Evolutionary Scenarios Explaining Levels of Diversity Observed in Virulence and Other Genes

*L. pneumophila* shows large genome plasticity and possesses a highly dynamic accessory genome due to its peculiar evolutionary conditions as an internal parasite of protozoa, its substantial horizontal gene transfer (HGT) and the presence of mobile genetic elements [12]. Mobile genetic elements together with bacteriophages and plasmids are called integrative conjugative elements [66]. Most of the *L. pneumophila* mobile genetic elements are integrative conjugative elements encoding different T4SS proteins which mainly consist of the *L. pneumophila vir* homologous regions (Lvh-region), the Trb/Tra family of conjugative elements and other genomic islands associated with the T4SS gene family. The Lvh-region consists of eleven genes encoding a T4SS and is located on DNA islands with high CG content [42,67]. The Lvh T4ASS is not present in the *L. pneumophila* strain Corby, but a similar T4ASS is integrated in this site (tmRNA), and a second genomic island carrying a T4ASS is integrated in the tRNA gene, a site which is not occupied by a mobile genetic element in the strains Lpn-Paris, Lpn-Lens or Lpn-Philadelphia [67].

In addition to HGT, gene duplication contributes to the evolution of bacterial genomes [59]. It is one of the main driving forces of genetic diversity, adaptation to specific environments and speciation. Four paralog genes (*sdeA, sdeB, sdeC* and *sdeE*) are an example of gene duplication in the expanded Dot/Icm effector repertoire [11,68]. Khodr et al. [59] summarized the distribution of integrative conjugative elements and essential duplication genes present in *L. pneumophila* strains. These genomic elements could be observed in our study in the isolates A129, A156, H29, H34, H35, A193, A195 and L04-545. In this study, most of the T4SS mobile genetic elements or duplication genes do not express a strong virulence phenotype.

According to David et al. [15], a certain local microevolution could be observed if *L. pneumophila* isolates from the same site of isolation have been obtained at different times. Such a case of microevolution could have occurred in the set of isolates obtained from the HZI hot water supply. Strain H23 of Gt14(30) had more time to diversify by genetic drift since it was isolated in 2013 in contrast to the isolates H1, H2, and H3, which were obtained in 2009. It is also possible that the *L. pneumophila* strain was subjected to different selection pressures during its life in the hot water supply system. If this is the case, then a likely explanation is that the harsh conditions in the hot water selected for changes in the metabolism of the isolates, making them able to adapt to this environment [69]. This selection process could render *L. pneumophila* more virulent because it adapted better to the high temperatures that occur in the human body. Differentiating between these putative evolutionary scenarios will be difficult and will require a greater understanding of the effect of diversity within hot water samples and isolates from this environment (Appendix A). To this end, further genomic and metagenomic studies are needed.

## 5. Conclusions

The selection of strains by MLVA and subsequent genomic characterization showed nearly the whole phylogenomic breadth of *L. pneumophila* as it was recovered within our study. SNP typing revealed a highly accurate typing superior to MLVA analysis.

## Figures and Tables

**Figure 1 microorganisms-11-00449-f001:**
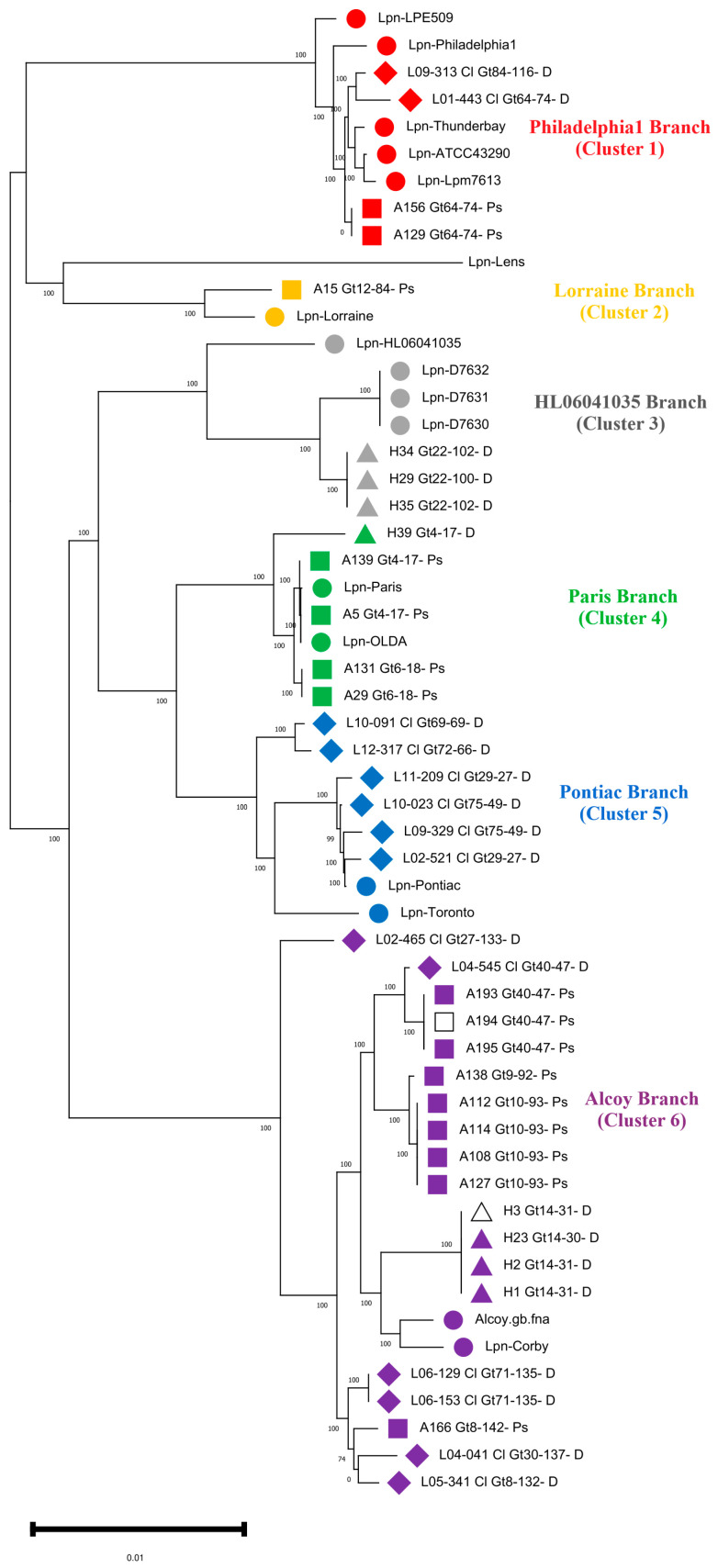
Phylogenomic tree based on Single Nucleotide Polymorphism (SNP) analysis of 55 *L. pneumophila* genomes. The strains group within six different clusters or branches marked in different colors. As a reference for this phylogenomic tree and to create the core genome *L. pneumophila,* strain Alcoy was used. Red color indicates isolates assigned to Philadelphia, orange indicates isolates assigned to Lorraine, gray indicates isolates assigned to HL060410035, green indicates isolates assigned to Paris, blue indicates isolates assigned to Pontiac and violet indicates isolates assigned to Alcoy branch. Reference genome from GenBank is indicated with filled symbols. Environmental isolates from the West Bank are depicted with filled squares. Environmental isolates from Germany (HZI) are indicated with filled triangles, and clinical isolates from Germany are indicated with filled rhomboids. The two isolates (A194_Gt40-47-PS and H3_Gt14-31-D) are indicated by an empty square and a triangle, respectively, both being sequenced using the PacBio platform.

**Figure 2 microorganisms-11-00449-f002:**
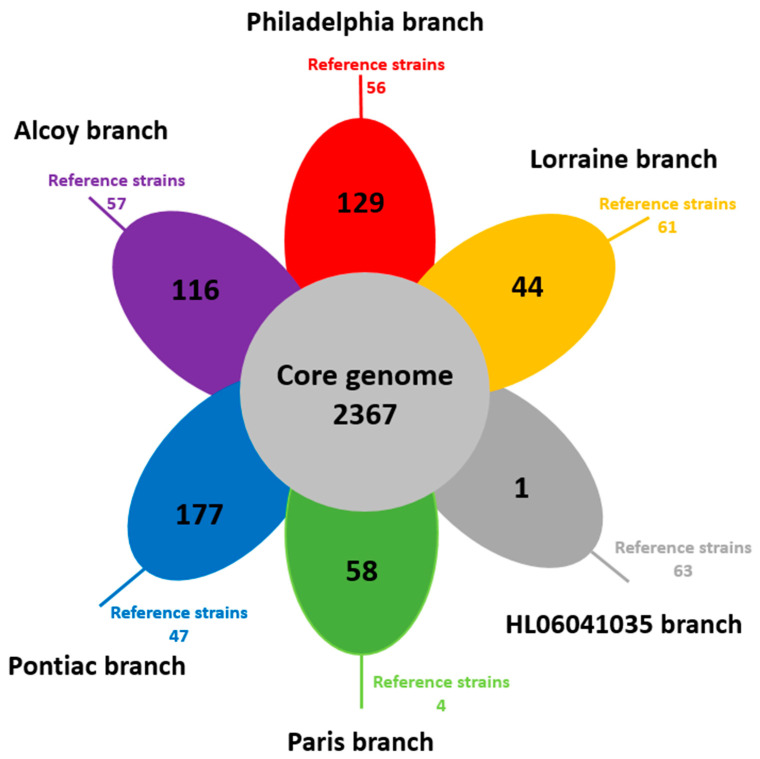
Venn diagram showing the shared and specific gene content of the six branches of *L. pneumophila* genomes. Each petal represents a specific branch with an associated color. The number in the center of the diagram represents the orthologous groups of genes shared by all the genomes. The number inside of each individual petal corresponds to the specific genes of each branch with non-orthologous genes in any of the other branch genomes. The number outside the petal represents the specific genes in the reference strains. Often, those genes could be assigned to plasmids.

**Figure 3 microorganisms-11-00449-f003:**
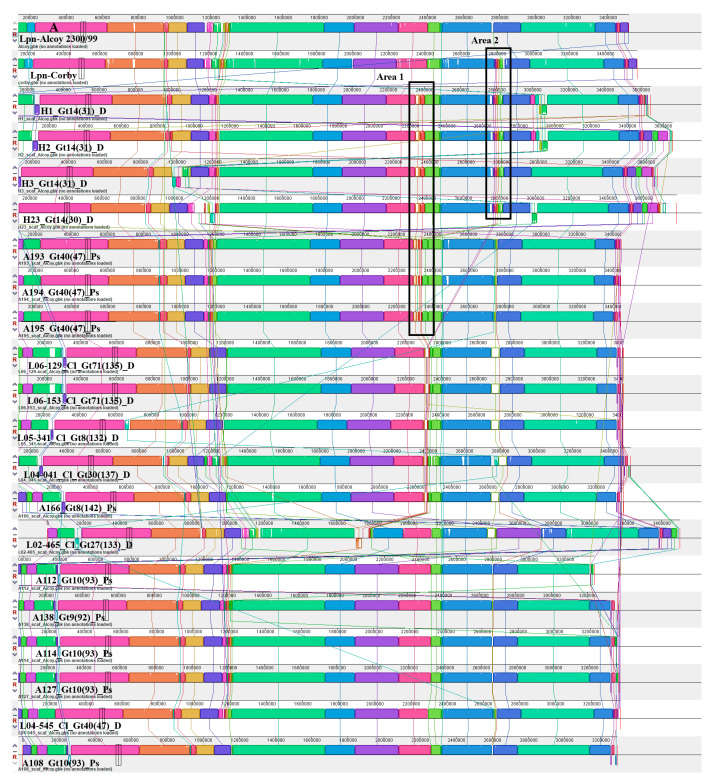
Mauve whole-genome alignments of *L. pneumophila* strains within Alcoy branch. (**A**) Progressive Mauve was used to compare the complete genomes of isolates assigned to Alcoy against strain Alcoy 2300/99 itself. Contigs have been ordered before by applying the “Move contigs” function. Contig borders are shown with red lines within the linear display. Boxes with common colors are pairwise-computed LCBs (locally collinear blocks). Mauve was run using default parameters, as described in (34 and 35). The cluster six organizations, as well as the identity and location of the ~84-kb area 1 and the ~49-kb area 2 elements, are shown. (**B**) Magnified in Mauve.

**Figure 4 microorganisms-11-00449-f004:**
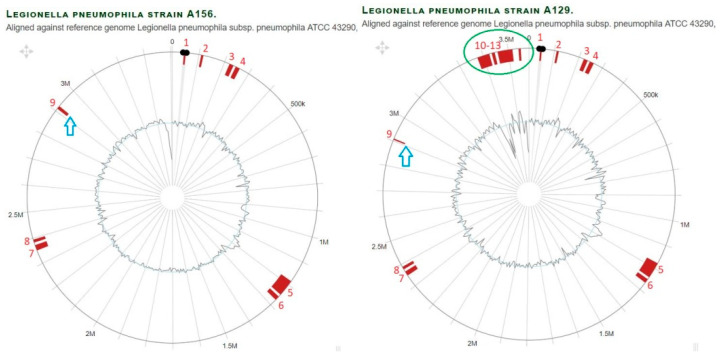
Comparison of genomic islands in branch Philadelphia of *L. pneumophila* strain A129 and A156 aligned against reference genome *L. pneumophila* strain Philadelphia1. With regard to the outer layers, the circle shows: (i) nucleotide positions in Megabase pairs (Mbp) (black); (ii) Island-Viewer-annotated potential genomic islands (GI) are labeled accordingly (red); (iii) extra GIs (marked by green oval); (iv) GI with different sizes (blue arrow).

**Figure 5 microorganisms-11-00449-f005:**
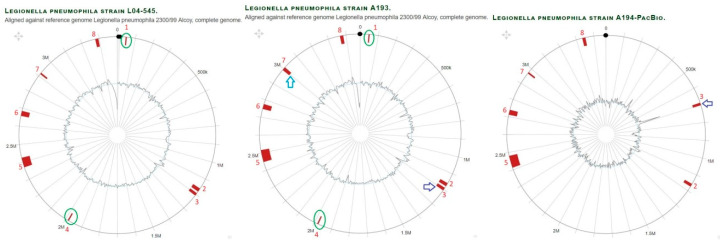
Comparison of genomic islands in branch Alcoy of *L. pneumophila* strains A193, A194 and L04-545 (also represents A195, not shown) aligned against reference genome *L. pneumophila* strain Alcoy 2300/99. With regard to the outer layers, the circle shows: (i) nucleotide positions in megabase pairs (Mbp) (black); (ii) Island-Viewer-annotated potential genomic islands (GI) are labeled accordingly (red); (iii) extra GIs (marked green oval); (iv) GI with different sizes (blue arrow); (v) GI translocation (purple arrow).

**Figure 6 microorganisms-11-00449-f006:**
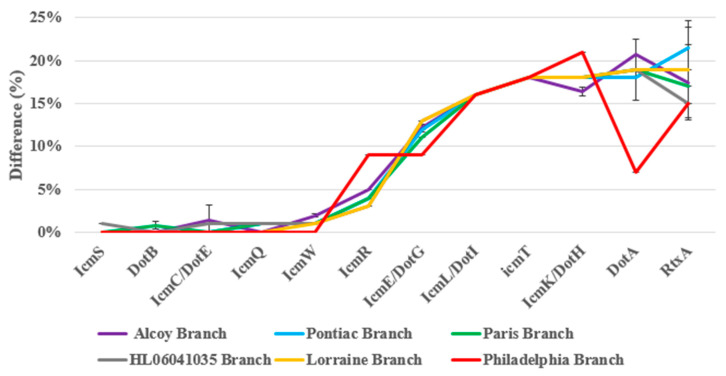
Amino acid differences of the fifty-five *L. pneumophila* strains genome sequenced in the pore-forming mediated cytotoxicity virulence genes, as identified by BLASTp search against the VFDB using *L. pneumophila* strain Philadelphia as the reference genome.

## Data Availability

Assembled genome sequence data are available at NCBI GenBank under BioProject accession number PRJNA912116. A detailed list of all GenBank accession numbers is provided in Appendix A. Isolate L10-023 has been reported before at GenBank under Bi-oProject accession number PRJNA278111.

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
