# Peer review of "Comparative Genomics of Legionella pneumophila Isolates from the West Bank and Germany Support Molecular Epidemiology of Legionnaires’ Disease"

_microorganisms, 2023, doi:10.3390/microorganisms11020449_

Round 1
Reviewer 1 Report
the article is well written, it brings new data about Legionella but it's a large paper and dificult to read.
sugestions:
1-adding all the tables in the supplementary file makes it easier to read.
2- reduce results description,
3-the paper has a lot of results but lacks a guiding line, it should be more structured.
4- miss conclusion.
Author Response
Comments and Suggestions for Authors
the article is well written, it brings new data about Legionella but it's a large paper and dificult to read.
sugestions:
1-adding all the tables in the supplementary file makes it easier to read.
We agree with the reviewer and moved all tables to supplemental material as essential values were mainly reported in the main text.
2- reduce results description,
We massively streamlined the presentation of the results, removed reports on hypothetical genes, redundancies with respect to outcome. Furthermore, we shortened the reports on genomic findings. Altogether, we reduced the results description from 316 to 270 lines.
3-the paper has a lot of results but lacks a guiding line, it should be more structured.
As of main importance for molecular epidemiology we started with general features, phylogenomic/SNP analysis, orthologs and report detailed on MLVA genotyping and its concordance to SNP analysis.
Subsequent reports based on genomic findings now have been shortened to be more focused.
4- miss conclusion.
We added a short and comprehensive conclusion based on our data.

Reviewer 2 Report
In this paper, L. pneumophila isolates were studied in two widely different geographical areas, i. e. West Bank and Germany. The whole genome of clinical and environmental isolates of L. pneumophila covering different MLVA-8(12) genotypes in the two areas was sequenced. Sequencing was conducted using Illumina HiSeq platform. In addition, two isolates (A194 and H3) were sequenced using a Pacific Biosciences (PacBio) RSII platform to generate complete reference genomes from each of the geographical areas. Genome sequences from L. pneumophila strains, including reference strains, were aligned with the genome sequence of the closest strain (L. pneumophila strain Alcoy). A whole genome phylogeny based on single nucleotide polymorphisms (SNPs) was created using the ParSNP software.
The work is well-organized and interesting, however, I recommend linguistic editing because some paragraphs are too long and some grammatical errors were noticed.
Author Response
Comments and Suggestions for Authors
In this paper, L. pneumophila isolates were studied in two widely different geographical areas, i. e. West Bank and Germany. The whole genome of clinical and environmental isolates of L. pneumophila covering different MLVA-8(12) genotypes in the two areas was sequenced. Sequencing was conducted using Illumina HiSeq platform. In addition, two isolates (A194 and H3) were sequenced using a Pacific Biosciences (PacBio) RSII platform to generate complete reference genomes from each of the geographical areas. Genome sequences from L. pneumophila strains, including reference strains, were aligned with the genome sequence of the closest strain (L. pneumophila strain Alcoy). A whole genome phylogeny based on single nucleotide polymorphisms (SNPs) was created using the ParSNP software.
The work is well-organized and interesting, however, I recommend linguistic editing because some paragraphs are too long and some grammatical errors were noticed.

Reviewer 3 Report
The study is very well planned and the manuscript is very well written. I have no specific comments. The article can be accepted for the publication.
Author Response
The study is very well planned and the manuscript is very well written. I have no specific comments. The article can be accepted for the publication.

Round 2
Reviewer 1 Report
I have no sugestions.